# OptBatch: Optimizing Instruction Tuning with Data Selection through Batch Stratified Sampling

## Abstract

Instruction tuning has optimized the specialized capabilities of large language models (LLMs), but it often requires extensive datasets and prolonged training times. The challenge lies in developing specific capabilities by identifying useful data and efficiently fine-tuning. High-quality and diverse pruned data can help models achieve lossless performance at a lower cost. In this paper, we propose **OptBatch**, a novel data selection method that focuses on the learnability of whole batch data rather than individual samples. OptBatch considers the coverage of the data distribution through stratified sampling and maximizes the relative distance between samples within a batch to enhance diversity. Furthermore, OptBatch utilizes Hessian gradient optimization to guide the selection strategy for subsequent batches. OptBatch effectively captures the intrinsic value of data curation, surpasses previous state-of-the-art methods, and demonstrates robust generalization performance across diverse downstream tasks and models. Extensive experiments reveal that OptBatch training in various pruning rates outperforms full dataset training, reducing computational cost by 20-40%. Additionally, evaluations using GPT-4 scores and other metrics for multi-turn dialogue, multilingual translation and QA tasks consistently demonstrate OptBatch's optimal performance.

## 1 Introduction

Instructional tuning is widely applied to enhance specific capabilities in LLM, including translation, dialogue, and domain-specific knowledge (Ouyang et al., 2022). Recent advances have prioritized the collection of high-quality data and the deployment of online training strategies (Hong et al., 2024; Xia et al., 2024a; Everaert & Potts, 2023). Despite the abundance of domain data, variability in quality remains a significant concern. In computer vision, data pruning techniques are utilized to maintain performance while reducing computational costs (Abbas et al., 2024; Tirumala et al., 2023; Abbas et al., 2023). In contrast, the presence of duplicate and low quality data can adversely affect performance in NLP (Hernandez et al., 2022). Therefore, identifying non-redundant and diverse data to optimize instruction tuning remains a critical and unresolved challenge.

Recent research highlights the importance of data diversity and coverage (Zheng et al., 2022; Hong et al., 2024; Zheng et al., 2024), categorizing approaches into online and offline methods. Offline methods leverage static features for importance scores, such as loss (Jiang et al., 2018; Wei et al., 2020), influence scores (Koh & Liang, 2017; Yang et al., 2022), or embedding-based metrics such as distance. These methods do not adapt scores based on model updates and necessitate pre-processing of large datasets. Online methods utilize importance scores calculated in real-time, such as those derived from reference models (Mindermann et al., 2022a; Deng et al., 2023) and gradients (Hong et al., 2024; Paul et al., 2021; Pooladzandi et al., 2022). Approaches based on a reference model for scoring require substantial foundational capabilities of the model (Mindermann et al., 2022a). Hong et al. (2024) aims to enhance orthogonal representativeness in online batch selection. However, their method emphasizes the directional diversity without considering the learnability of the data.

In this paper, we focus on employing stratified sampling and maximizing intersample distances for online batch selection to enhance sample diversity. We define high-quality, untrained examples as "learnable samples", aiming to identify these in each batch to enhance model training. Using stratified sampling, we consider samples within various score ranges in a batch, rather than focusing solely on high-score samples. Methods that prioritize high-loss may be overly influenced by outliers and noisy data, potentially compromising the model robustness (Lyu & Tsang, 2019; Xia et al.,

2022; Karamcheti et al., 2021). To increase diversity, we maximize the relative distance between data within a batch. Samples with similar importance scores may share characteristics, leading to a lower gain in new information. These redundant samples lack learnability (Hong et al., 2024). Given the differing distributions between batches, we draw inspiration from the Adam (Kingma & Ba, 2017) optimization algorithm to integrate second-moment cumulative gradient updates. This approach mitigates fluctuations due to the random sampling of different batches, helping the model to consistently recognize critical features between batches.

Our approach emphasizes data diversity, learnability, and real-time feedback, exhibiting the following properties: (1) We propose an online loss-probability based stratified sampling algorithm that prioritizes batch selection methods with higher diveristy. (2) We utilize Hessian gradient optimization to guide the data selection strategy for the next batch. The diversity we emphasize is rooted in gradient norms, and we demonstrate that the gradients are Lipschitz continuous. (3) We evaluate our approach on three diverse downstream datasets: llama3-Chinese-chat (LLaMaQA), WikiMatrix and our net literature dialogue dataset (NetLit). Our results demonstrate that OptBatch consistently achieves optimal loss across various pruning rates and models, including LLaMa3 and ChatGLM3. This indicates its capacity to maintain the same loss at a reduced computational cost.

## 2 PRELIMINARIES

### 2.1 GRADIENT NORM

To effectively select relevant samples from our dataset, we represent each sample using features, denoted as $F$. Common representations of $F$ utilize embeddings (Sorscher et al., 2022; Zheng et al., 2022; Xia et al., 2022). However, embeddings capture only the intrinsic features of the samples and do not reflect the model's importance during training. Instead, we employ gradient representations (Xia et al., 2024b; Wang et al., 2023a) because gradients provide a dynamic measure of sample relevance, capturing each sample's influence on model updates and learning efficacy.

To minimize redundant computational costs, we specifically compute the gradients from the `lm_head` layer. The gradients from this layer are defined as follows:

$$\text{gradient}_{\text{lmhead}} = \nabla l_{\text{output}}(z; \theta^t) \cdot h_{\text{lastlayer}}^T = (y_{\text{pred}} - y_{\text{label}}) \cdot h_{\text{lastlayer}}^T, \tag{1}$$

$$\text{gradient}_{\text{norm}} = \|\text{gradient}_{\text{lmhead}}\|_{2,\text{axis}=1} \tag{2}$$

where $\text{gradient}_{\text{lmhead}} \in \mathbb{R}^{D \times H}$. $D$ denotes the vocabulary size of the model, and $H$ represents the dimension of the last hidden layer's output. The term $\nabla l_{\text{output}}(z; \theta^t)$ signifies the gradient of the output layer with respect to the model parameters $\theta^t$. The transpose of the last layer's hidden states is given by $h_{\text{lastlayer}}^T$. Additionally, $\text{gradient}_{\text{norm}}$ quantifies the magnitude of the gradient from the `lm_head` layer using the $L_2$ norm along axis 1.

To mitigate the computational overhead associated with token-level gradient calculations, we adopt a sequence-level gradient approach. We leverage the gradient from the final layer to represent the entire sequence. The gradient at the last layer captures highly abstracted features of the input data, effectively reflecting the overarching trends and significant characteristics of the sequence.

### 2.2 ADAPTIVE MOMENT ESTIMATION

The Adam (Kingma & Ba, 2017) algorithm utilizes exponentially weighted moving averages to estimate the momentum and the second moment of the gradients, defined by the following variables:

$$\mathbf{v}_t \leftarrow \beta_1 \mathbf{v}_{t-1} + (1 - \beta_1)\mathbf{g}_t, \tag{3}$$

$$\mathbf{s}_t \leftarrow \beta_2 \mathbf{s}_{t-1} + (1 - \beta_2)\mathbf{g}_t^2, \tag{4}$$

where $\beta_1 = 0.9$, $\beta_2 = 0.999$, $\mathbf{g}_t^2$ represents the element-wise square of the gradient. It allows the variance estimate to evolve more slowly than the momentum estimate, promoting stable convergence during optimization. To correct for bias in the estimates, Adam employs the following formulas to normalize the momentum and variance estimates, and to rescale the gradients for parameter updates:

$$\hat{\mathbf{v}}_t = \frac{\mathbf{v}_t}{1 - \beta_1^t}, \quad \hat{\mathbf{s}}_t = \frac{\mathbf{s}_t}{1 - \beta_2^t}, \tag{5}$$

$$\mathbf{H}_t = \frac{\hat{\mathbf{v}}_t}{\sqrt{\hat{\mathbf{s}}_t} + \epsilon}, \quad \theta_t \leftarrow \theta_{t-1} - \eta \mathbf{H}_t. \tag{6}$$

where $\eta$ denotes the learning rate and $\epsilon = 1e - 6$. The normalized estimates $\hat{\mathbf{v}}_t$ and $\hat{\mathbf{s}}_t$ help compute the adaptive learning signal $\mathbf{g}'_t$, guiding the model's parameter updates.

We propose substituting the gradient norm with the Hessian gradient $H_t$ to address significant distributional discrepancies among samples. By leveraging historical gradient information, $H_t$ effectively captures global features, thereby enhancing the stability of gradient updates during training. This modification alleviates the instability that may arise due to the idiosyncratic nature of specific batches, promoting consistency of gradients between batches throughout the optimization process.

## 3 METHOD

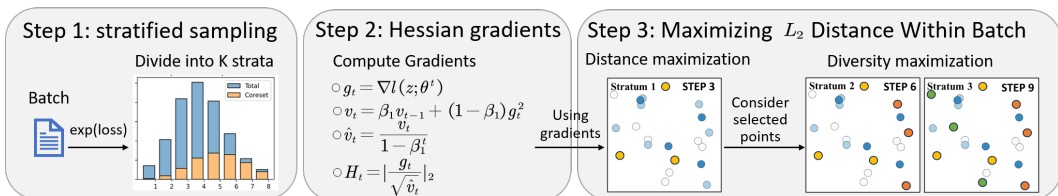

Figure 1: **The workflow of OptBatch. Step 1:** We divide a batch of data into $K$ strata based on loss. Then, we select $|S|$ data according to the probability of $\exp(\text{loss})$ and calculate the number of data in each stratum. **Step 2:** We calculate the Hessian gradient $H_t$ of the data as features. **Step 3:** For stratum 1, we randomly initialize a point. We calculate the distance to the first point and select the farthest point from the same stratum as the second point. We update the minimum distance from the remaining points to the selected point and repeatedly choose $|S_i|$ samples. For strata 2 and 3, in order to select points for the new stratum, we need to consider the selected points from the previous strata. Finally, we will obtain a diversified subset that is relatively far from each other.

As illustrated in Figure 1, OptBatch initially computes the loss for each individual sample through forward propagation and employs stratified sampling to determine the number of samples in each stratum. Specifically, we select $n_i$ samples from each stratum with the objective of maximizing the distance between the Hessian gradients of the selected samples. This approach establishes a theoretical upper bound for the training loss of the chosen subset. In subsection 3.1, we demonstrate the existence of this upper bound and extend the geometric properties to encompass the training gradients. In subsection 3.2, we detail the incorporation of prior gradient updates during the computation of new batches and utilize Hessian approximations to enhance the precision of gradient magnitude evaluations throughout the optimization process. In subsection 3.3, we perform stratified sampling for each batch of data, selecting samples by maximizing the $L_2$ distance.

### 3.1 LIPSCHITZ CONTINUITY OF THE GRADIENT

Introducing the geometric set cover into the dataset spatial distribution to select the coreset with the maximum coverage can ensure that coreset still follows the original distribution $P$ at a high pruning rate. The dataset can be represented as $S = \{(x_i, y_i)\}_{i=1}^N$, where $x_i = (x_i^1, x_i^2, \ldots, x_i^T)$ is the input token sequence and $y_i$ is the prediction sequence $Y = (y_t^1, y_t^2, \ldots, y_t^T)$. The loss function $l$ is defined as $l_{CE} = -\frac{1}{T} \sum_{t=1}^T \log p(y^t = x^t | x^{<t})$. The goal of coreset selection $S'$ is to remove redundant and unimportant data while minimizing the model's loss under a given pruning rate $\alpha$: $\min_{S' \subset S: \frac{|S'|}{|S|} \leq 1-\alpha} \mathbb{E}_{x,y \sim P} [l(x, y, h_{S'})].$

Analyzing the training process of a model on an $r$-cover coreset reveals that it incurs a bounded risk for the dataset. (Sener & Savarese, 2017; Zheng et al., 2022) prove that the loss function is Lipschitz continuous and bounded, enabling loss-based stratified sampling. Gradients can quantify inter-sample distances and gauge the coverage of selected samples, as they more accurately reflect sample differences and importance. Maximizing gradient distances between samples within the subset thus enhances the coverage of the entire dataset. Consequently, we demonstrate that gradients

exhibit Lipschitz continuity and are constrained by an upper bound. The proof of gradient Lipschitz continuous can be found in the Appendix A.

$$\|\nabla l(x, y; h'_S)\| \leq rL_s + \sqrt{\frac{L^2 \log\left(\frac{1}{\gamma}\right)}{2n}} \tag{7}$$

### 3.2 Hessian-approximated Gradient Optimization

To integrate global information, OptBatch builds upon the Adam optimizer (Kingma & Ba, 2017) and leverages a Hessian-approximated gradient optimization to account for variations in gradient curvature across batches. Specifically, we focus on the gradient $\mathbf{g}_t = \nabla l(z; \theta^t)$ derived from a batch sample. The updates for the second moment estimate $\hat{\mathbf{v}}_t$ are defined as Equation 5. Hessian gradients $H_t$ are derived from the norm of the gradient, adjusted by the second moment:

$$H_t = \left\| \frac{\mathbf{g}_t}{\sqrt{\hat{\mathbf{v}}_t}} \right\|_{2, \text{axis}=1} \tag{8}$$

where $H_t \in \mathbb{R}^D$, $D$ denotes the vocabulary size of the model. Significant distributional discrepancies among batches may result in anomalous behavior. To capture global gradient characteristics effectively, we substitute $H_t$ for the original gradient norm. Refer to subsection 2.2 for details.

---

**Algorithm 1:** OptBatch

---

**Input:** Batch $B$; current model $f$; pruning rate $\alpha$; the number of strata $k$; Hessian gradient $H_t$.
**Output:** coreset $S$

1 loss $\leftarrow f_\theta(B)$; coreset size $|S| = \alpha * |B|$;
2 coreset $S \leftarrow$ Sample $|S|$ samples without replacement according to the probability exp(loss);
3 $B_0, B_1, ..., B_{k-1} \leftarrow$ Split loss in $\mathbb{B}$ into k ranges with an even range width;
4 $S_0, S_1, ..., S_{k-1} \leftarrow$ Split loss in $\mathbb{S}$ into k ranges with an even range width;
5 $S_{00} \leftarrow$ Randomly select one point in $B_0$;
6 **for** *i in range(k)* **do**
7     $distance \leftarrow \min(L_2$ distances of $B_i$ to selected points $S$, axis $= 0$); // using $H_t$ feature
8     **for** *j in range($|S_i|$)* **do**
9        $S_{ij} \leftarrow \arg\max(distance)$;
10        $distance' \leftarrow$ Calculate the distances of the remaining points to $S_{ij}$;
11        $distance \leftarrow \min(distance, distance')$;
12     **end**
13 **end**
14 **return** *coreset* $S$;

---

### 3.3 Optimizing Online Batch Selection

In each training step, we access a data batch $B$ and aim to select a representative subset of samples $S$. Data selection should consider the subset as a cohesive entity rather than as isolated samples; this approach effectively mitigates redundancy and addresses potential omissions within the dataset, thereby promoting greater diversity (Hong et al., 2024). When selecting a data point, the exclusion of similar points is crucial, as it enhances the informational value available for model training.

**Step 1: Stratified sampling based on loss probability.** To ensure that our sampling aligns with the overall distribution of the data while accounting for both challenging and easy samples, we employ a loss probability based stratification approach. This method is detailed in algorithm 1. We partition the batch into $K$ strata based on loss values, utilizing the approach outlined by CCS (Zheng et al., 2022). The width of each stratum is determined by the formula (max_loss − min_loss)/$K$. Subsequently, we calculate the size of the selected sample set as $|S| = |B| \times \alpha$, where $\alpha$ denotes the pruning rate. In our sampling process, we use exp(loss) as the selection probability for each sample, and we sample from the dataset without replacement to create the initial set. These samples are then categorized into $K$ strata, allowing us to tally the number of samples in each stratum.

**Step 2: Maximizing $L_2$ Hessian gradient distance within the batch.** In each stratum, we aim to maximize the $L_2$ Hessian gradient distance among points to ensure greater separation within

the batch as illustrated in Figure 2. Specifically, for the number of points to be sampled in each stratum, indicated by $S_i$, we begin by randomly selecting one point. Subsequently, we compute the distances from the remaining points in the stratum to this selected point, referred to as $distance$. We then identify the point that is farthest away as the second point. We continue by calculating the $distance'$ from the remaining points to this second point, updating distance as $distance = min(distance, distance')$. This iterative process continues until we have selected the required number of points, as specified by $S_i$.

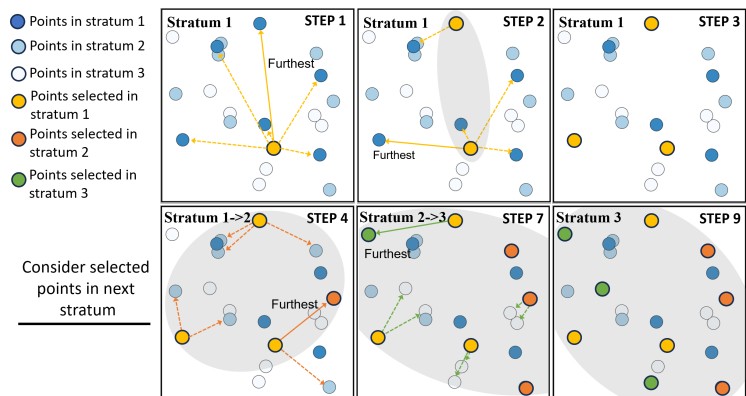

Figure 2: Maximizing $L_2$ Distance Within the Batch. Step 1: For stratum 1, we randomly initialize a point and calculate the $L_2$ distance from the points in the same stratum to the first point. Step 2-3: We select the point that is farthest from the first point as the second point. We then update the minimum distance from the remaining points to the selected points and iteratively choose $|S_i|$ (e.g. 3) samples. Steps 4 and 7: For stratum 2, we need to consider the selected points from the previous strata. We calculate the minimum distance from the points in stratum 2 to the selected points and take the maximum one as the first point. We then iterate to complete the selection for stratum 2. Step 9: Finally, we obtain a subset that is relatively far apart and diverse within the batch.

## 4 EXPERIMENT

### 4.1 EXPERIMENTAL SETUP

**Datasets.** We conduct our finetuning on the following instruction tuning datasets: (1) Net Literature Dialogue Dataset (NetLit); (2) llama3-Chinese-chat (LLaMaQA) [1]; (3) WikiMatrix[2] (Schwenk et al., 2019). NetLit consists of multi-turn dialogues related to web literature. Each entry is 1000 characters long, with totally 100 million training examples. LLaMaQA is the dataset for the Chinese version of LLaMa3, which includes system instructions, queries, and GPT-4 responses, amounting to 1.69 million training samples. WikiMatrix extracts parallel sentences in all possible language pairs from the content of Wikipedia. We use 10 language pairs with 28 million training examples.

**Implementation details.** We train with two base models: Meta-LLama-3-8B-Instruct (AI@Meta, 2024) and ChatGLM-3-6B (GLM et al., 2024). The hyper-parameters we used for full-fine tuning are as follows. We fine-tune the base model for 1 epoch with AdamW using a cosine learning rate scheduling strategy, $\beta_1 = 0.9$, $\beta_2 = 0.999$, $\epsilon = 1e - 8$. We set the initial learning rate is set to $1e-5$. The batch size is set to 32, the context window's maximum length is 2048 tokens, and longer examples are trimmed to fit in. We use $16 \times$ A800 80G GPUs for training.

**Baselines.** We compare our **OptBatch** with various baseline methods, including random selection, online hard, CCS (Zheng et al., 2022), InfoBatch (Qin et al., 2023). **Random selection** randomly select data from training datasets. **Online hard** starts by selecting data points with the highest loss and continues downwards until enough data points are chosen. **CCS** stratifies the data based on loss and randomly selects a fixed number of data points from each stratum. It utilizes important scores derived from the computer vision domain; however, we substitute it with loss. **InfoBatch**

---

[1]https://modelscope.cn/datasets/baicai003/Llama3-Chinese-dataset

[2]https://github.com/facebookresearch/LASER/tree/main/tasks/WikiMatrix

randomly removes samples with low information content based on the loss distribution, then adjusts the gradients of the remaining samples to match the original gradients. The threshold is set to the average loss. Nevertheless, this approach proves inadequate when dealing with high pruning rates. To address this limitation, we increase the threshold appropriately for higher pruning rates.

## 4.2 MAIN RESULTS

### 4.2.1 OPTBATCH PERFORMANCE COMPARISON

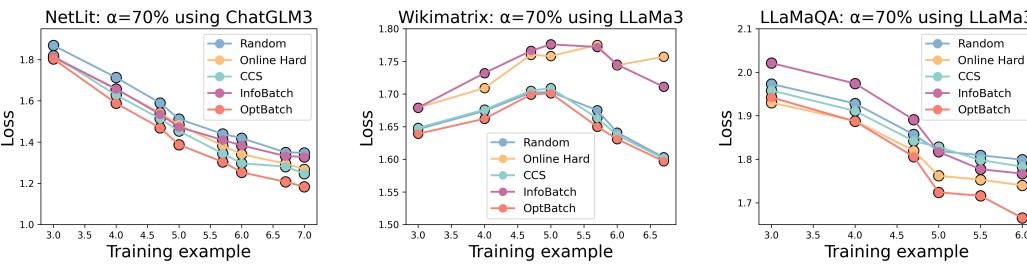

Figure 3: Evaluation on different datasets using ChatGLM3 model with pruning rate 70%

**Performance under different datasets.** Based on the comparisons in Figure 3, we have the following observations: (1) Our findings indicate that OptBatch exhibits superior performance across a variety of datasets. In particular, both the NetLit dataset and the WikiMatrix translation dataset reveal that OptBatch consistently achieves the lowest loss across all evaluated datasets. Notably, while OptBatch initially underperforms compared to InfoBatch at smaller data sizes within the LLaMaQA dataset, it ultimately surpasses InfoBatch as the data size increases. (2) OptBatch strikes a balance between diversity coverage and task difficulty, thereby achieving enhanced performance across all three downstream tasks. The performance of CCS and InfoBatch varies significantly across different datasets, presenting a contrasting trend. CCS demonstrates superior efficacy compared to InfoBatch in both the NetLit and WikiMatrix datasets, suggesting that the aspect of diversity coverage is particularly crucial within the realms of web literature and translation tasks. Conversely, the LLaMaQA dataset places a greater emphasis on the dimension of task difficulty.

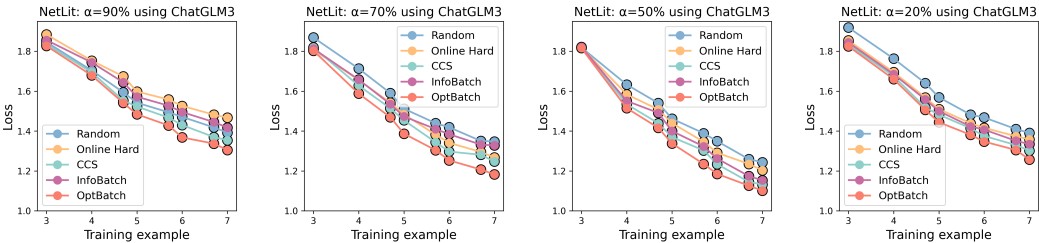

Figure 4: Evaluation with different pruning rates on NetLit using ChatGLM3 model

**Performance under different pruning rates.** In Figure 4, we have the following observations: (1) OptBatch consistently demonstrates the lowest loss across various pruning rates. At high pruning rate, InfoBatch likely retains the hard data while pruning easy data, resulting in increased losses akin to Online Hard. (2) As illustrated in Figure 6, the loss decreases consistently from $\alpha = 20\%$ to its minimum at $\alpha = 50\%$ with OptBatch, indicating a potential 50% reduction in computational costs. Although the loss at $\alpha = 70\%$ is marginally higher than at $\alpha = 50\%$, it yields a 20% reduction in computational costs. Notably, the losses at $\alpha = 90\%$ and $\alpha = 20\%$ are remarkably similar, likely due to the high redundancy in the web text domain. Thus, selecting a representative 10% of the data can achieve nearly the same loss as using 90% of the data, which includes redundant information. These findings suggest that the selection of a batch characterized by diversity and informativeness can markedly enhance the model's performance while concurrently reducing computational costs.

**Performance between different models.** In Figure 5, we find that OptBatch exhibits superior stability across various models, consistently achieving the lowest loss values in both cases exam-

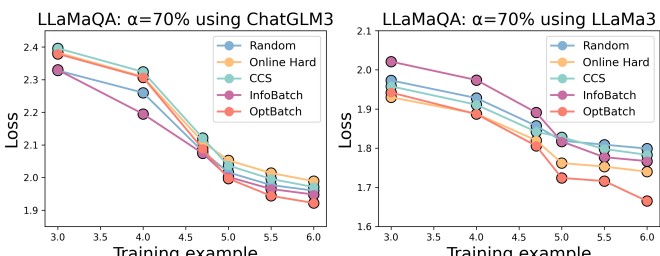 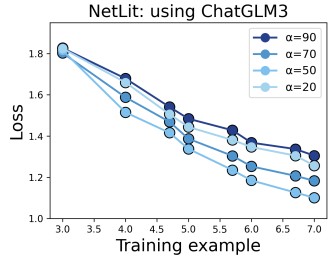

Figure 5: Evaluation on LLaMaQA using ChatGLM3 and LLaMa3 with pruning rate 70%

Figure 6: OptBatch under different pruning rates

ined. In contrast, InfoBatch and CCS demonstrate varying degrees of instability across the same models. Specifically, OptBatch is particularly effective for the ChatGLM3 model when applied to large datasets, while InfoBatch is more advantageous for smaller datasets. Conversely, the LLaMa3 model shows a stronger alignment with the Online Hard and OptBatch methodologies.

**Feature selection.** In Figure 9, we employ embedding, gradient norm, and hessian gradient as features and conduct experiments on the NetLit dataset with an 70% pruning rate. Our findings indicate that the Hessian gradient achieve the highest performance, suggesting that optimizing the gradient direction is more effective than relying on static features such as embedding. Additionally, we should consider global gradient consistency to enhance the representativeness of the data subset.

## 4.3 EVALUATION

**GPT-4 Evaluation.** For our net literature dataset NetLit, we are more concerned with the performance of LLMs to generate responses under various pruning methods, similar to the capabilities of LLMs in role-playing scenarios (Shao et al., 2023; Wang et al., 2023b). We evaluate the performance on personality and speaking style as the character's primary feature. We instruct GPT-4 to score the generated responses, and the Chain of Thought (Wei et al., 2022) process allows us to evaluate the performance of different pruning methods effectively. We referred to the prompt template (Shao et al., 2023). GPT-4 scores each response from 1 to 5, with 5 indicating strong alignment with the character's personality and speaking style, and 1 indicating poor alignment. As illustrated in Figure 7(a), OptBatch has the highest percentage of high-score examples, reaching 60.5%. In comparison, the CCS method achieves 52.6%, while the InfoBatch method stands at 43.5%.

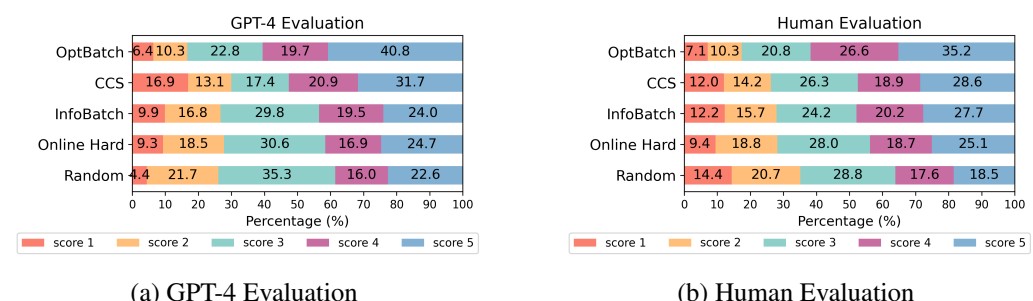

(a) GPT-4 Evaluation

(b) Human Evaluation

Figure 7: Scores of generated responses in terms of GPT-4 evaluation and human evaluation

**Human Evaluation.** Furthermore, human judgment is still the most thorough and realistic assessment of whether the generated response is character-aligned. Some poor GPT-4 annotation cases are discovered during our task. We invite annotators to rectify the scoring results of GPT-4 for each test data, leading to human evaluation results. Detailed prompts for response generation and more cases can be found in Appendix B. As shown in the Figure 7(b), OptBatch achieves 61.8% in the high-score range, compared to 47.5% for CCS and 47.9% for InfoBatch. Additionally, OptBatch produces fewer low-score examples than the other methods.

Table 1: Reference-based Metrics Evaluation on LLaMaQA using LLaMa3 and ChatGLM3

| LLaMa3(ChatGLM3) | Random | Online Hard | InfoBatch | CCS | OptBatch |
|---|---|---|---|---|---|
| Bleu-4 | 22.09 (13.40) | 27.04 (10.43) | 25.31 (13.16) | 26.17 (13.03) | **27.11 (13.73)** |
| Rouge-1 | 40.92 (34.82) | 43.52 (28.10) | 40.73 (35.02) | 42.57 (34.60) | **44.07 (35.30)** |
| Rouge-2 | 21.79 (18.10) | 22.74 (11.86) | 22.62 (18.05) | 22.33 (17.71) | **22.81 (18.25)** |
| Rouge-L | 35.38 (30.03) | 38.77 (24.11) | 37.88 (30.22) | 37.31 (30.05) | **39.52 (30.53)** |

Table 2: Reference-based Metrics Evaluation on WikiMatrix(LLaMa3).

| | Random | Online Hard | InfoBatch | CCS | OptBatch |
|---|---|---|---|---|---|
| Bleu-4 | 31.52 | 24.46 | 26.76 | 32.24 | **32.94** |
| Rouge-1 | 44.57 | 39.62 | 42.67 | 46.17 | **47.51** |
| Rouge-2 | 21.69 | 13.84 | 19.45 | 22.56 | **22.84** |
| Rouge-L | 36.56 | 30.22 | 34.17 | 38.24 | **38.97** |

**Reference-based Metrics.** For the translation dataset WikiMatrix and question-answer dataset LLaMaQA, we employ industry-recognized Bleu-4(Papineni et al., 2002), Rouge-1, Rouge-2, and Rouge-L(Lin, 2004) as our metrics to validate the relevance. Due to LLaMa3's superior support for multilingual capabilities, we exclusively use the LLaMa3 model for training and validation on the WikiMatrix dataset Table 2. For the LLaMaQA dataset, we conduct training and validation using both LLaMa3 and ChatGLM3 Table 1. All experiments are conducted under an 70% pruning rate. We can observe that our OptBatch outperforms other methods across all 4 metrics, further demonstrating its universality across different tasks and models.

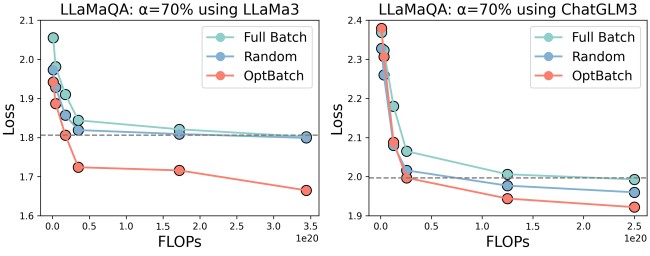
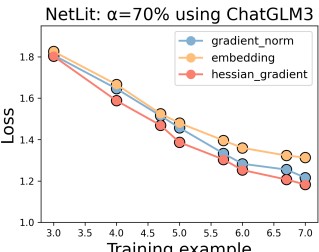

Figure 8: Comparison of FLOPs for Pruning and Full Data

Figure 9: Feature seletion

### 4.4 Efficient Performance

To quantify the computational savings achieved by our method, we compare the FLOPs (Floating Point Operations) required for both full batch data and pruned batch data using OptBatch. For full batch processing, the computation is represented by the formula:

$$\text{Full-Batch FLOPs} = B \times (L_i + L_o) \times F_f + B \times L_o \times F_b \tag{9}$$

where $B$ is the batch size, $L_i$ is the input sentence length, $L_o$ is the output sentence length, $F_f$ denotes the FLOPs for the forward pass of one token, and $F_b$ represents the FLOPs for the backward pass, which generally requires approximately twice the FLOPs of the forward pass. In the case of OptBatch, which incorporates pruning, the computational requirement is reduced by a factor corresponding to the pruning rate $\alpha$. The computation for OptBatch can be expressed as:

$$\text{OptBatch FLOPs} = B \times (L_i + L_o) \times F_f + (1 - \alpha) \times B \times L_o \times F_b \tag{10}$$

The backward pass computation is scaled down by $(1 - \alpha)$. For example, with an 80% pruning rate, only 20% of the backward pass computations need to be performed. The comparison of the computational requirements of OptBatch, Random, and Full Batch methods at an 70% pruning rate is illustrated in Figure 8. It is evident that our method can reduce computational requirements by at least 30% while maintaining equivalent loss. This demonstrates the efficiency of OptBatch in significantly lowering the computation burden without compromising the performance.

## 5 RELATED WORK

**Coreset selection.** (Guo et al., 2022; Yoon et al., 2021) aims to create a smaller subset of the original data that captures essential patterns for effective model training. Most pruning methods prioritize the selection of the most challenging samples (Sorscher et al., 2022). However, in large datasets, focusing on difficult samples may lead to the inclusion of outliers and noisy data. In contrast, Xia et al. (2022) selects samples near the median, thereby discarding redundant data while minimizing exposure to noise. Nonetheless, this approach does not account for the overall distribution and lacks diversity. Zheng et al. (2022) stratifies data into K strata based on importance scores and then selects an average number of samples from each stratum. However, this method aggregates and randomly samples from each cluster, failing to ensure that every selected sample contributes meaningfully to learning and compromising the overall data distribution.

**Online batch selection.** Online batch selection employs a subset of data in each batch, thereby accelerating model training. Online Batch selection is classified into two approaches, dependent on the reference model (Evans et al., 2024; Mindermann et al., 2022b) and not dependent (Hong et al., 2024; Qin et al., 2023). Hong et al. (2024) maximizes the orthogonal representativeness of the subset for online batch selection, yet focuses solely on directional diversity without accounting for the learnability of data points. Qin et al. (2023) achieves lossless training acceleration by pruning low-scoring samples and amplifying the gradients of remaining samples; however, their reliance on mean thresholds and pruning rates imposes limitations. OptBatch addresses these challenges by stratifying loss to consider the learnability of points across various intervals while ensuring diversity by controlling the relative distances among points.

**Feature Selection.** In image processing, embeddings are commonly employed to assess sample similarity (Sorscher et al., 2022; Zheng et al., 2022; Xia et al., 2022). However, embeddings solely reflect intrinsic data features and do not directly indicate a sample's contribution to model training. They also inadequately capture dynamic model changes during training and may neglect adversarial features or noise effects. In contrast, gradients effectively capture dynamic model changes throughout training and evaluate sample importance directly. Xia et al. (2024b) employ LoRA for warmup training, constructing a reusable gradient datastore of low-dimensional features. Wang et al. (2023a) propose a gradient-based method for influence estimation that eliminates Hessian inversion.

## 6 CONCLUSION

In conclusion, our proposed method **OptBatch**, significantly enhances the efficiency of instruction tuning for large language models (LLMs) by focusing on the learnability of whole batch data rather than individual samples. By employing stratified sampling to ensure data distribution coverage and maximizing the relative distance between batch samples for diversity, OptBatch effectively curates high-quality, diverse data. Additionally, it utilizes Hessian gradient optimization to guide batch selection strategy, leading to superior performance compared to previous state-of-the-art methods. Our experiments demonstrate that OptBatch achieves robust generalization across various downstream tasks and models, reduces computational cost by 20-40%, and consistently delivers optimal results in multilingual translation, QA datasets, and multi-dialogue evaluations.

**Limitations.** (1) $lm\_head$ gradients is not enough. We focus on the gradient in final layer to optimize computational resources. However, it proves inadequate for long sequences. We will investigate more effective methods to leverage sequence gradients while preserving computational efficiency. (2) Balancing difficulty and diversity. While our method achieves a balance between diversity and difficulty, difficulty assessment necessitates adaptation to specific contexts ( Appendix D). For example, during the fine-tuning of the LLaMa3 model on the OpenOrca dataset ( Appendix C), prior exposure to certain data may enhance the efficacy of the online hard method. In the future, we aim to dynamically adjust the difficulty ratio in response to situational demands. (3) Loss as the primary metric. Our main experiments primarily evaluate the model's performance by observing the decrease in loss. In reality, loss is not the only metric. In the future, we will incorporate the model's performance on various downstream tasks' accuracy as an additional evaluation metric.

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

## A  THEORETICAL ANALYSIS

Our proof flow is similar to the proof of Theorem 1 in (Sener & Savarese, 2017; Zheng et al., 2022). In previous studies, given n $i.i.d.$ samples drawn form $P_\mu$ as $S = (x_i, y_i)_{i\in[n]}$ where $y_i \in [\mathcal{C}]$ is the class label for example $x_i$. A coreset $S'$ which is a p-partial r-cover for $P\mu$ on the input space. The loss function $l(x, y, h'_S)$ is $\lambda_l$-Lipschitz continuous for all y, w and bounded by L, and the class-specific regression function $z(x; h'_S) = p(y = c \mid x)$ is $\lambda_\eta$-Lipschitz continuous for all c, and $l(x, y; h'_S) = 0, \forall (x, y) \in S'$. Then use Hoeffding's Bound and conclude that with probability at least $1 - \gamma$:

$$\left| \frac{1}{n} \sum_{x,y \in \mathcal{S}} l\left(x, y; h_{\mathcal{S}'}\right) \right| \leq r\left(\lambda_l + \lambda_\eta LC\right) + \sqrt{\frac{L^2 \log\left(\frac{1}{\gamma}\right)}{2n}}$$

This formula is derived from the following:

$$\mathbb{E}_{y_i \sim \eta(x_i)}\left[l(x, y, h'_S)\right] \leq r\left(\lambda_l + \lambda_\eta LC\right)$$

According to the above Theorem, the loss function $l(x, y, h'_S)$ is $\lambda_l$-Lipschitz continuous. There is a constant $L_l > 0$ such that:

$$\|l(x, y, h'_S) - l(\hat{x}, \hat{y}, h'_S)\| \leq L_l\|x - \hat{x}\|$$

When a constant $L_s$ is present, the gradient of the differentiable loss function $l(x, y, h'_S)$ is also Lipschitz continuous:

$$\|\nabla l(x, y; h'_S) - \nabla l(\hat{x}, \hat{y}; h'_S)\| \leq L_s\|x - \hat{x}\|$$

*Proof*

Assuming that the Hassian matrix $H(x)$ of the loss function $l(x, y, h'_S)$ is bounded, it means that there exists a constant $M$ such that for all x:

$$\|H(x)\| \leq M$$

For $\hat{x}$ in $r$ ball around $x$, the gradient can be further represented using Talor expansion:

$$\nabla l(\hat{x}, \hat{y}; h'_S) = \nabla l(x, y; h'_S) + H(x)(\hat{x} - x) + o(\|\hat{x} - x\|)$$

where $o(\|\hat{x} - x\|)$ represents an infinitely small quantity of higher order.

The difference of gradient is estimated as:

$$\|\nabla l(\hat{x}, \hat{y}; h'_S) - \nabla l(x, y; h'_S)\| = \|H(x)(\hat{x} - x) + o(\|\hat{x} - x\|)\|$$

According to the triangle inequality, we can get:

$$\|\nabla l(\hat{x}, \hat{y}; h'_S) - \nabla l(x, y; h'_S)\| \leq \|H(x)(\hat{x} - x)\| + \|o(\|\hat{x} - x\|)\|$$

Since $H(x)$ is bounded, we can get:

$$H(x)\|\hat{x} - x\| \leq \|H(x)\|\|\hat{x} - x\| \leq M\|\hat{x} - x\|$$

For $o(\|\hat{x} - x\|)$, we can find a constant $k$ such taht for a sufficiently small $\|\hat{x} - x\|$:

$$\|o(\|\hat{x} - x\|)\| \leq k\|\hat{x} - x\|$$

To sum up, we can get:

$$\|\nabla l(\hat{x}, \hat{y}; h'_S) - \nabla l(x, y; h'_S)\| \leq (M + k)\|\hat{x} - x\|$$

Let $L_s = M + K$, then the gradient Lipschitz continuous is satisfied.

We further use the Hoeffding's Bound and conclude that with probability at least $1 - \gamma$:

$$\|\nabla l(x, y; h'_S)\| \leq rL_s + \sqrt{\frac{L^2 \log\left(\frac{1}{\gamma}\right)}{2n}}$$

So we can get the conclusion that if the Hessian matrix of loss function is bounded, its gradient is Lipschitz continuous. This property is very important in optimization and algorithm analysis because it guarantees the convergence and stability of the algorithm.

## B  MORE DETAILS ON PROMPT CONSTRUCTION AND DIALOGUE CASES

Table 3: Prompt for GPT4 to evaluate Personality.

| **Prompt Template(Chinese — *English*, /\*...\*/indicates that some context is omitted)** |
| --- |
| 你将得到一个由人工智能助手模仿角色{character_name}所写的回答。你的任务是使用给定的评价标准，按照评估步骤对{character_name}的回答进行打分。以下是数据： │ *You will be given responses written by an AI assistant mimicking the character {character_name}. Your task is to rate the performance of {character_name} using the specific criterion by following the evaluation steps. Below is the data:* 

 【角色人设】 │ 【*Character Profile*】 
 /\*...\*/ 


 【历史对话】 │ 【*Historical Dialogue*】 
 /\*...\*/ |

**Table 3 – continued from previous page**

**Prompt Template(Chinese — *English*, /\*...\*/indicates that some context is omitted)**

【User的当前讲话】｜【*User's Current Speech*】

/\*...\*/

【角色的参考回答】｜【*The Reference Response for character*】

/\*...\*/

【角色的回答】｜【*The Response for character*】

/\*...\*/

【评估标准】｜【*Evaluation Criterion*】

1-5分：【角色的回答】是否能够反映角色的人设、个性、说话风格。｜*Scores 1-5: Whether*【*The Response for character*】*reflects the character's character profile, personality, and speaking style.*

【评估步骤】｜【*Evaluation Steps*】

1.阅读【角色人设】，总结角色的个性和特征。｜*1.Read*【*Character Profile*】*and summarize the character's personality and characteristics.*

2.阅读【历史对话】，识别角色的个性和说话风格。｜*2.Read*【*Historical Dialogue*】*and summarize the character's personality and speaking style.*

3.在对以上有清晰的理解后，阅读【User的当前讲话】和【角色的回答】，将【角色的回答】与【角色人设】进行比较，分析一致性和不一致性，如果一致则反映【角色的回答】是符合角色的个性和说话风格。｜*3.After having a clear understanding of the above, read*【*User's Current Speech*】*and*【*The Response for character*】*, compare*【*The Response for character*】*to the*【*Character Profile*】*. Look for any consistencies or inconsistencies. Does the response reflect the character's personalities and speaking style?*

4.将【角色的回答】和【角色的参考回答】进行比较，分析相似和不相似，如果相似则反映【{character_name}的回答】是符合角色的个性和说话风格。｜*4.Compare*【*The Response for character*】*to the*【*The Reference Response for character*】*. Look for any similar or not similar. Does the response reflect the character's personalities and speaking style?*

5.使用1到5分对【角色的回答】进行打分，评估【角色的回答】在多大程度上反映了角色的个性和说话风格。1表示完全不反映角色的个性和说话风格，5表示完美反映角色的个性和说话风格。｜*5.Use the given scale from 1-5 to rate how well the*【*The Response for character*】*reflects the personalities and speaking style of the character. 1 being not reflective of the character's personalities and speaking style, and 5 being perfectly reflective of the character's personalities and speaking style.*

【输出格式】｜【*Output Format*】

/\*...\*/

Continued on next page

**Table 3 – continued from previous page**

| Prompt Template(Chinese — *English*, /\*...\*/indicates that some context is omitted) |
|---|
| 首先以逐步的方式写出你对标准的理解，以确保你的结论是正确的。避免一开始就简单的陈述正确答案。然后在新的一行打印出与正确答案相对应的分数。最后，在新的一行上重复仅选中的分数。请严格按照【输出格式】的样式打分。｜ *First, write out in a step-by-step manner your reasoning about the criterion to be sure that your conclusion is correct. Then print the score on its own line corresponding to the correct answer. At the end, repeat just the selected score again by itself on a new line. Please score strictly according to the style of* 【*Output Format*】. |

Table 4: Case of celebrity character responses generated from Chatglm3 by different pruning methods. /\*...\*/indicates that some context is omitted.

| | Character Personality | Score |
|---|---|---|
| Character Profile | Elon Musk, the CEO of SpaceX and Tesla, and he is an entrepreneur full of passion and imagination. | |
| Context | /\*......\*/ | |
| User | What contribution have you made to the development of science and technology? | |
| Chatglm3(Random) | I've made a lot of contributions in tech, like developing AI, neural networks, and self-driving technology. | 2 |
| Chatglm3(Online Hard) | I think a lot of people know that I founded two companies: SpaceX and Tesla. I've also invested in many other innovative startups. As for the specific contributions I've made? I doubt many know because those contributions are really thanks to all the employees and partners. I'm just a leader guiding them toward success. | 3 |
| Chatglm3(CCS) | For example, the "Starlink" project I started lets people everywhere connect to the internet, even in remote areas. Plus, the "Starship" program aims to make humanity a truly multi-planetary species. | 4 |
| Chatglm3(OptBatch) | I really think we need to ditch fossil fuels as soon as possible, which is why I founded Tesla. By making electric cars, I hope to change people's lives and cut down global greenhouse gas emissions. Plus, I want to help make the dream of getting humans to Mars a reality. | 5 |
| User | What achievements in your life are worth mentioning? | |
| Chatglm3(Random) | I'm really proud of founding Tesla and SpaceX. Tesla is the greenest electric car out there. Our aim is to make humanity a multi-planet species. We're all about pushing tech forward and getting ready for an amazing future! | 4 |
| Chatglm3(Online Hard) | I've managed to build reusable rockets, which is seen as one of the coolest tech achievements of the 20th century. I really believe that soon, we're going to make interstellar travel happen. | 2 |

**Table 4 – continued from previous page**

| | Character Personality | Score |
|---|---|---|
| Chatglm3(CCS) | Most of my achievements are in entrepreneurship. I started with Zip2, then created PayPal, and I also acquired X.com and Square. In 2008, I launched SpaceX, where we cut rocket costs by 70%. Then in 2019, I launched a bunch of cool companies like Neuralink and The Boring Company, plus a few others focused on energy and innovation. | 4 |
| Chatglm3(OptBatch) | I'm really proud of a few things I've done. In 2018, Tesla became one of the first companies to mass-produce and popularize electric cars, which really helps with cutting down pollution. Our solar tech is also top-notch in the industry. And let's be real, we're making strides toward getting humans to Mars, even though we've still got a long way to go. | 5 |

Table 5: Case of virtual character responses generated from Chatglm3 by different pruning methods. /*...*/indicates that some context is omitted.

| | Character Personality | Score |
|---|---|---|
| Character Profile | 九爷傅宸，商业总裁之子，傅氏集团继承人之一，人称九爷，高冷霸道。 | |
| Context | /*......*/ | |
| User | 我之前一直都不敢向你表明我的心意，今天真的是我鼓足了勇气才和你说的... | |
| Chatglm3(Random) | (傅宸挑了挑眉)陆小姐，勇气可嘉，但时机不对。 | 3 |
| Chatglm3(Online Hard) | (傅宸语气冷漠)你有什么事情要说吗，我还有工作要做。 | 3 |
| Chatglm3(CCS) | (傅宸看着你，眉头紧皱)说吧，你到底想表达什么？ | 4 |
| Chatglm3(OptBatch) | (傅宸冷笑一声，眼神中带着一丝嘲讽)你以为我会相信吗？别装可怜了。 | 5 |
| User | 不是所有的感情都是为了利益，我只是想和你分享我的想法。 | |
| Chatglm3(Random) | (傅宸勾起嘴角)在我眼里，想法可不是轻易能被接受的。你有什么让我关注的理由？ | 3 |
| Chatglm3(Online Hard) | (傅宸皱了皱眉，语气阴沉)在这个圈子里，真心往往被利益掩盖。你真的认为我会轻易相信吗？ | 4 |
| Chatglm3(CCS) | (傅宸起身，眼神阴沉)在我这个位置，信任可不是轻易得到的。你有什么让我在意的理由？ | 4 |
| Chatglm3(OptBatch) | (傅宸冷哼一声，眼神冷冽)分享想法？在这个世界上，真诚常常被算计。你觉得我会轻易相信你的话吗？ | 5 |

## C THE PERFORMANCE IN OPENORCA DATASET

OpenOrca (Lian et al., 2023) is based on Google's Flan dataset, which includes system instructions, queries, and GPT-4 responses, amounting to 4.21 million training samples. We utilize the

LLaMa3 model and train it on OpenOrca using various methodologies. The potential pre-exposure of LLaMa3 to the dataset may account for the consistently low loss values and the negligible differences in the effectiveness of the different training approaches.

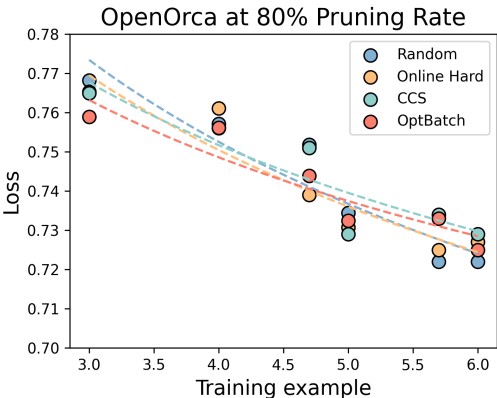

Figure 10: Evaluation on OpenOrca using LLaMa3 model with pruning rate 80%

## D  DISTRIBUTION OF DIFFERENT DATA SELECTION STRATEGIES

Hard sampling prioritizes the selection of samples starting from the most challenging (highest scoring) ones. Conversely, CCS emphasizes coverage, ensuring nearly equal representation of samples across different score ranges. OptBatch integrates these approaches, balancing the consideration of both difficult and simple samples. This strategy prevents the model from experiencing excessive initial difficulty, thereby avoiding a significant increase in loss.

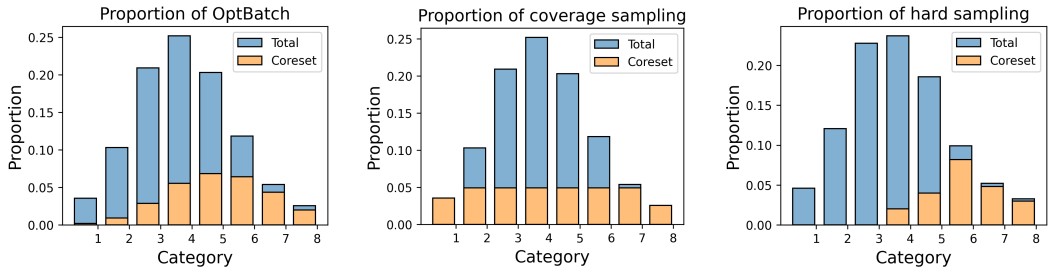

Figure 11: The data distribution under OptBatch, CCS and hard sampling

