# OpenReview forum: "OptBatch: Optimizing Instruction Tuning with Data Selection through Batch Stratified Sampling"
_ICLR.cc/2025/Conference — ICLR 2025 Conference Withdrawn Submission_

### Official Review · Reviewer_FCys · 2024-10-15

**Soundness:** 2
**Presentation:** 2
**Contribution:** 2
**Rating:** 3
**Confidence:** 4

**Summary:**

This paper proposes OptBatch for instruction-tuning data selection. The core idea is to take the Hessian gradient as a datapoint's feature and select datapoints by maximizing the distances within their features. To let OptBatch account for both challenging and easy samples, it stratifies datapoints based on loss and calculates each stratum's selection size. The paper showed that OptBatch outperforms several baselines on LLaMaQA, WikiMatrix, and NetLit.

**Strengths:**

+ I appreciate the novelty of the idea.
+ I personally believe it is important to study data selection methods that consider the diversity of the selected datapoints. In my understanding, if we take the gradient-based feature as "how much new information the datapoint can bring to the model", OptBatch maximizes "new information" by diversifying the gradient-based features of selected datapoints - this motivation makes a lot of sense to me.

**Weaknesses:**

+ **Evaluations are poor**: In the modern setup of instruction tuning, the loss is NOT indicative of the real performance at all. Commonly, people take only the GPT/Human win-rate (or score) as the metrics for measuring performance (it is kind of a standard way these days).
  + Most evaluations in the paper take loss as the metrics (in both main and auxiliary experiments), and the paper does GPT/Human win-rate (or score) evaluation on ONLY NetLit.
  + In my understanding, LLaMaQA is a standard **open-ended** instruction-response dataset (like WildChat, ShareGPT, Alpaca, etc), so the best way to do evaluation is still GPT/Human win-rate (or score) evaluation instead of metrics like BLEU or ROUGE.
  + Overall, the paper's evaluation is poor and unreliable IMO, which cannot validate the real effectiveness of OptBatch.
+ **Important design choices are unjustified**: An important design choice of OptBatch is stratifying datapoints based on their losses. Please feel free to point it out if I missed anything.
  + Why do you do that? I saw `methods that prioritize high-loss may be overly influenced` but it is does not explain why you adopt this design choice.
  + There is also no ablation study to prove its necessity.
  + What's the number of K? I think it is an important hyperparameter and even some experimental analysis is needed (at least K=1 is relevant to the ablation study mentioned above).
+ **Presentation could be improved (not a major concern)**
  + The paper could try to make some motivations and intuitions more clear. For example, The reason why `we use exp(loss) as the selection probability` (line 212) is unclear.
  + In Algorithm 1, the notation usage is confusing.
    + Why is there $\mathbb{B}$ and $\mathbb{S}$ in lines 3 and 4, which didn't exist before?
    + It seems that you want to first do some sampling to fix the selected datapoint number in each stratum, so $S$ in line 2&4 is different from line 9&14. If my understanding is correct, it would be better to use different notations. It is very confusing to let $S$ contain both selected datapoints and datapoints in the preliminary sampling process (just used to fix the number of selected datapoints).

My current overall score for this paper is 3, which is below the acceptance threshold. However, I would be happy to consider increasing my score if the authors can address (even part of) my concerns.

**Questions:**

+ Where is the reference or link for the NetLit dataset? Or is it a private dataset? Sorry if I missed any detailed description about NetLit in the paper.
+ Some papers appeared more than once in the REFERENCES section, e.g., *Less: Selecting influential data for targeted instruction tuning*.
+ How do the authors think about other ways of maximizing the distances within selected embeddings/vectors? One example could be Diversify and Conquer: [Diversity-Centric Data Selection with Iterative Refinement](https://arxiv.org/abs/2409.11378).

---

### Official Review · Reviewer_MWgR · 2024-11-02

**Soundness:** 3
**Presentation:** 2
**Contribution:** 2
**Rating:** 3
**Confidence:** 4

**Summary:**

The paper proposes OptBatch, a data selection method designed to optimize instruction tuning for large language models (LLMs) by focusing on whole-batch data learnability. The approach uses stratified sampling to ensure coverage of the data distribution, maximizing inter-sample diversity within batches by increasing relative distances between samples. Additionally, it employs Hessian gradient optimization to guide the selection strategy for subsequent batches, enhancing generalization and reducing computational cost by 20-40% without sacrificing model performance. Experiments across tasks such as multilingual translation, dialogue, and question answering show that OptBatch outperforms prior methods, achieving lower loss and improved computational efficiency.

**Strengths:**

- **Originality**: The paper introduces a novel data selection technique, OptBatch, which leverages stratified sampling combined with Hessian gradient optimization to maximize diversity and learnability in batch selection.
- **Quality**: The authors thoroughly evaluate OptBatch against several baseline methods. Experimental results across multiple datasets demonstrate that OptBatch achieves competitive or superior performance while significantly reducing computational costs.
- **Clarity**: The paper is generally well-organized and structured, presenting the motivation, methodology, and experimental results in a clear sequence.
- **Significance**: OptBatch addresses a crucial challenge in instruction tuning for LLMs by reducing training data volume without compromising model accuracy, making it impactful for applications requiring cost-effective scaling.

**Weaknesses:**

1. **Unclear Notation**:
The paper suffers from unclear and inconsistent notation, which hampers understanding of the key equations and methods:

    - In Equation (1), the notation for $l$ lacks clarity in definition. It’s unclear how $l$ is computed, leaving the derivation and insight behind Equation (1) and (2) unclear.
    - In Equation (6), the introduction of the Hessian gradient is confusing. Adam is generally known as a first-order optimization method, so the addition of a Hessian term deviates from standard practice and isn’t adequately justified. Furthermore, the paper inconsistently switches between bold and italic symbols around Equation (6), which adds unnecessary confusion.

2. **Usage of Equation (7)**:
Equation (7) is insufficiently integrated into the proposed method. The paper fails to clarify how this equation aligns with the broader methodology, making its inclusion feel confusing and extraneous within the current context. More detailed explanations are needed to make this equation’s purpose clear to readers.

3. **Hyperparameter $k$ (Number of Strata)**:
The paper lacks a discussion on the hyperparameter $k$, which determines the number of strata. There is no mention of how $k$ is set or how it impacts the performance of OptBatch. Providing insight into $k$’s influence on performance would help clarify how stratified sampling affects the results and guide practical implementation.

4. **Lack of Novelty**:
The method is not particularly novel. Similar coreset selection strategies based on clustering have been introduced in prior works, such as in [1], [2] and [3]. The similarity of OptBatch to these clustering-based coreset methods raises questions about the incremental contribution of this approach.

[1] TAGCOS: Task-agnostic Gradient Clustered Coreset Selection for Instruction Tuning Data

[2] GRAD-MATCH: Gradient Matching based Data Subset Selection for Efficient Deep Model Training

[3] Deep Batch Active Learning by Diverse, Uncertain Gradient Lower Bounds

**Questions:**

Please refer to Weaknesses.

**Details Of Ethics Concerns:**

I am concerned about the human annotators' background and how they are tasked in the human judgement.

---

### Official Review · Reviewer_q8yK · 2024-11-02

**Soundness:** 4
**Presentation:** 1
**Contribution:** 3
**Rating:** 6
**Confidence:** 4

**Summary:**

The paper proposes an online data selection method for instruction tuning of LLMs that combines difficulty and diversity. To ensure the difficulty they apply stratified sampling with the probability proportional to the exponential of loss. To ensure the diversity, they greedily sample examples to maximize Hessian gradient distance to existing samples. Experimental results show that the method can achieve lower loss under different pruning rates and better response quality under both GPT-4 and human evaluations.

**Strengths:**

1. The method combines difficulty and diversity to select samples. It is more reasonable than methods can consider only one aspect.
2. They propose to calculate the distance with Hessian gradient. Ablation study in Figure 9 shows that Hessian gradient is better than embedding or gradient norm when calculating the distance.
3. To measure the response quality, they provide both GPT-4 and human evaluations. They also provide the loss under different computes in Figure 8 to show that the proposed method can achieve the most computational savings among different methods.

**Weaknesses:**

1. The settings of human evaluation (the number and qualification of human annotators, instructions given to them, etc) are missing. From the given details, the human annotators are provided the scoring results of GPT-4, which might weaken the value of human evaluation.
2. The writing of the paper is poor. Especially the theoretical analysis in Section 3.1 is confusing. I am not sure whether it really supports any part of the algorithm. The authors should consider moving the subsection after the algorithm description and clearly stating which step the theoretical analysis supports.

**Questions:**

1. Is the loss in Figure 3, 4, 5, 6, 8, 9 training loss or validation loss? If it is the validation loss, how is the validation set built for each dataset? Also, what is the unit of training example in the x-axis?
2. Appendix C mentions that Openorca is not suitable due to the potential pre-exposure of Llama3 to the dataset. Can this be addressed by using the base model of Llama3 instead of the instruction model?

---

### Official Review · Reviewer_PB22 · 2024-11-04

**Soundness:** 2
**Presentation:** 2
**Contribution:** 2
**Rating:** 5
**Confidence:** 3

**Summary:**

This paper introduce a novel data selection method for instruction tuning named OptBatch. OptBatch propose an online loss-probability based stratified sampling algorithm to select batch with higher diversity and use hessian gradient optimization to guide next batch selection. Experiments on multilingual translation, QA datasets, and multi-dialogue conversations shows that OptBatch can maintain the same loss at a reduced computational cost.

**Strengths:**

1. Consistently outperform other baselines across different models, datasets and pruning rates.
2. Fair computational saving (~30%) to maintaining equivalent loss.
2. Human judgment for more robust evaluation.

**Weaknesses:**

1. All the experiments is conducted for one training epoch. Is one epoch best for final performance?
2. It's unclear whether OptBatch will finally reach similar loss compared to full data training. If not, OptBatch may not suitable for practical use because we may perfer FLOPs saving under the same performance.

**Questions:**

1. In most of the figures, what does the "Training Samples" means?
3. What the time complexity of the online batch selection procedure?

---

### Note · Authors · 2024-11-13

**Comment:**

Dear Editor,
	I would like to express my sincere gratitude to the hardworking staff of your conference for the efforts in reviewing our manuscript. I am deeply sorry to submit a request for the withdrawal our paper to your esteemed conference. We believe that there are some areas in the manuscript that require further improvement, and out of our sense of responsibility towards your conference, we have decided to withdraw our paper after careful deliberation and discussion.
	We understand that this decision may cause inconvenience, and we deeply apologize for any waste of time and resources that may have been incurred. We want to assure you that this decision was made in the best interest of maintaining the scientific integrity of them manuscript, and we do not wish to publish work that does not meet the highest standards of quality. Once again, we sincerely apologize for any inconvenience caused to your conference. Thank you for your understanding and cooperation in this matter.

**Withdrawal Confirmation:**

I have read and agree with the venue's withdrawal policy on behalf of myself and my co-authors.